# Trends in Myocardial Infarction Morbidity and Mortality from Ischemic Heart Disease in Middle-Aged Lithuanian Population from 2000 to 2023: Data from Population-Based Kaunas Ischemic Heart Disease Register

**DOI:** 10.3390/medicina61050910

**Published:** 2025-05-17

**Authors:** Ricardas Radisauskas, Lolita Sileikiene, Daina Kranciukaite-Butylkiniene, Sarunas Augustis, Erika Jasukaitiene, Dalia Luksiene, Abdonas Tamosiunas, Karolina Marcinkeviciene, Dalia Virviciute, Diana Zaliaduonyte, Gintare Sakalyte

**Affiliations:** 1Institute of Cardiology, Medical Academy, Lithuanian University of Health Sciences, LT-50162 Kaunas, Lithuania; lolita.sileikiene@lsmu.lt (L.S.); daina.butylkiniene@lsmu.lt (D.K.-B.); sarunas.augustis@lsmu.lt (S.A.); erika.jasukaitiene@lsmu.lt (E.J.); dalia.luksiene@lsmu.lt (D.L.); abdonas.tamosiunas@lsmu.lt (A.T.); karolina.marcinkeviciene@lsmu.lt (K.M.); dalia.virviciute@lsmu.lt (D.V.); gintare.sakalyte@lsmu.lt (G.S.); 2Department of Cardiology, Medical Academy, Lithuanian University of Health Sciences, LT-50161 Kaunas, Lithuania; diana.zaliaduonyte@lsmu.lt

**Keywords:** acute myocardial infarction, ischemic heart disease, morbidity, mortality, trends

## Abstract

*Background and Objectives*: Over the past decades, various epidemiological analyses have reported a significant decrease in the number of deaths related to cardiovascular diseases (CVDs). Trends in acute myocardial infarction (AMI) morbidity and mortality from ischemic heart disease (IHD) were less studied in Eastern and Central Europe. This study aimed to determine and evaluate changes in AMI morbidity and mortality from IHD among the middle-aged urban Lithuanian population during 2000–2023. *Materials and Methods*: The data source was the Kaunas ischemic heart disease registry for residents aged 25–64. The diagnosis of AMI was based on the proposed epidemiological criteria used in the WHO MONICA project protocol. Age-standardized morbidity and mortality rates were calculated per 100,000 population. The changes in morbidity and mortality rates were calculated using the Joinpoint regression analysis method, and changes presented as a percentage estimate per year. *Results*: During 2000–2023, it was observed that age-standardized AMI morbidity significantly changed in the 25–64-year-old male and female population (−1.3%/yr., *p* = 0.006 and −2.3%/yr., *p* < 0.001, respectively). In males aged 25–54, a significant decrease in AMI morbidity rates by an average of 2.2%/yr. (*p* < 0.001) was found, contrary to the males aged 55–64, where morbidity was without substantial changes. We found a significant decrease in AMI morbidity in both age groups (the younger and older) of females, by 2.1%/yr. (*p* = 0.002) and 2.4%/yr. (*p* < 0.001), respectively. In the 25–64-year-old male population mortality from IHD significantly decreased (−2.0%/yr., *p* < 0.001), whereas in females it did not significantly change. Mortality from IHD in males aged 25–54 and 55–64 years significantly decreased by an average of 3.3%/yr. (*p* = 0.002) and 1.2%/yr., (*p* = 0.004), respectively. No significant trends in mortality from IHD in both age groups of females over the past 24 years were observed. *Conclusions:* During the study period, the age-standardized AMI morbidity among Kaunas middle-aged males and females significantly decreased. The age-standardized mortality from IHD decreased significantly among Kaunas middle-aged males, but there were no significant changes among females.

## 1. Introduction

Over the past decades, various epidemiological analyses have reported a significant decrease in the number of deaths related to cardiovascular diseases (CVDs) in most European Union (EU) countries [1,2,3]. Among CVDs, the morbidity of acute myocardial infarction (AMI) and mortality from ischemic heart disease (IHD) remain the main causes of CVD [4,5]. Most of the previous analyses carried out in Western and Northern EU countries found that the rates of morbidity of AMI and mortality from IHD have significantly decreased [6,7,8,9,10,11,12,13]. Eastern and Central EU countries have been less frequently studied for trends on AMI morbidity and mortality from IHD, and the results have been very heterogeneous [14,15,16,17]. Eastern and Central European countries, including Lithuania, have historically been under-represented in CVD research, particularly in large-scale multinational studies. Several factors contribute to this gap. One of the main barriers is limited data infrastructure and interoperability, which hinders contributions to or access of large CVD registries. Differences in healthcare system organization, such as differences in emergency response methods, advanced interventions, and follow-up care models, also complicate cross-country comparisons and discourage standardized data collection. In recent years, the COVID-19 pandemic may have introduced some corrections to the changes in the morbidity of AMI and mortality from IHD. In most studies, the incidence of AMI and mortality from IHD have been on a decreasing trend, and, in particular, in-hospital mortality from AMI has been decreasing in some European countries [18,19]. In contrast, in-hospital mortality has been stable in other European countries over the past decade [14,17]. These trends were mainly associated with more advanced diagnostics and treatment of acute coronary syndromes using the latest technologies [20]. The possibility of timely application of primary and secondary prevention of IHD, by adjusting and controlling the levels of risk factors, plays an equally important role [21,22]. Studies have also identified different rates of AMI morbidity and mortality from IHD in sex and age groups in individual European countries.

The region of Eastern and Central European countries, which also includes the Baltic countries, stands out with higher and more stable rates of AMI morbidity and mortality from IHD. The rates of AMI morbidity and mortality from IHD in the Baltic countries, particularly in Lithuania, stand out as somewhat higher compared to both Eastern, Central, and Western European countries. Meanwhile, in recent years, the rates of IHD mortality in the Baltic countries have been decreasing, but the rates of morbidity of AMI have remained stable. Recent epidemiological studies have shown that the mortality from IHD has decreased in many Central and Eastern European countries over the past decades [3]. During 2012–2020, standardized mortality rates from AMI for males remained unchanged in some Central and Eastern European countries such as Bulgaria, Romania, Slovakia, and Slovenia, while they significantly decreased in Poland, Hungary, the Czech Republic, Latvia, Estonia, and Croatia. During the corresponding period, standardized mortality rates from AMI for females remained unchanged in Bulgaria, Croatia, Slovakia, and Slovenia, while they significantly decreased in Hungary, the Czech Republic, Estonia, Latvia, Poland, and Romania [3].

Studies have shown that this decrease is due to several factors, such as improved management of risk factors for CVD. High blood pressure, high cholesterol, and smoking have been relatively successfully controlled in many Central and Eastern European countries [23,24,25]. The increased use of preventive medicines, such as statins and antiplatelet agents, to control dyslipidemia and prevent blood clots has also contributed to the decrease in the incidence of AMI. Improved access to and care for acute coronary syndromes (ACSs) through faster coronary artery disease screening and removal have also contributed to the decreasing trends in mortality from IHD in recent years [18].

Despite intensive efforts to control the incidence of ACS, in Lithuania, the incidence of AMI has been stable over the past decade, but the mortality from IHD has tended to decrease [11]. However, both the incidence of AMI and mortality from IHD remain about two times higher on average compared to some Central and Eastern European countries [26]. Of concern is the three–four times higher rates of both AMI morbidity and mortality from IHD among middle-aged Lithuanian males compared to females, as well as compared to other Central and Eastern European countries.

This study aimed to determine and evaluate changes in the morbidity of AMI and mortality from IHD among Lithuanian middle-aged urban persons from 2000 to 2023 based on data from the Kaunas ischemic heart disease registry.

## 2. Materials and Methods

### 2.1. Study Sample

The data source is the Kaunas ischemic heart disease registry for residents aged 25–64. Information on Kaunas’ population aged 25–64 years from 2000 to 2023 was obtained from the State Data Agency [27]. The diagnosis of AMI and quality control procedures were based on the proposed epidemiological criteria according to the WHO MONICA project protocol [28], which has been described in detail elsewhere [29]. The study used the “Cold pursuit” technique (i.e., retrospective data collection) after the subject had already been discharged from the healthcare institution or had died. The following documentation was used to identify and verify cases of possible AMI or death from IHD (hospital discharge statistical forms, patient medical histories, outpatient medical histories, autopsy and forensic medical protocols, and death certificate information). For all cases, a possible AMI was identified and confirmed and a special questionnaire was filled out according to the WHO MONICA protocol form “AMI Event Registration Form”. The epidemiological diagnostic category (EDC) of possible AMI or possible coronary death was defined by the following four diagnostic criteria: (1) symptoms of a coronary event, (2) dynamic changes in the electrocardiogram (ECG) indicating the development of AMI, (3) changes in blood enzyme activity, and (4) autopsy findings [28]. Based on the above EDC, all possible AMI cases were divided into the corresponding four EDCs as “definite AMI”, “possible AMI” (or “possible coronary death” in the case of death), “no AMI” (or the cause of death was not coronary artery disease), and “insufficient data to define the category”. Cases with the EDCs “definite AMI” and “possible AMI (for survivors) and “possible coronary death” and “insufficient data to define AMI” for deceased cases were included in the study. Each AMI event had to be evident during the study period and had to occur more than 28 days after any previously recorded AMI event. Multiple AMI attacks occurring within 28 days of the onset of first symptoms were considered a single event. An AMI event was defined as fatal if death occurred within the first 28 days of the onset of the illness. If the patient survived beyond 28 days of the onset of the attack, the AMI cases were classified as non-fatal.

Throughout the study period, the same methods for AMI case identification, the same diagnostic criteria of AMI, and the same quality assurance procedures were applied to ensure the comparability of the data. All data were evaluated by socio-demographic variables such as sex and age. The study also assessed data by the final clinical diagnosis of events using codes of the International Classification of Diseases 10 modification (ICD-10) (codes I21–I22 and unstable angina pectoris code I20.0) and EDC classification (“definite AMI” and “possible AMI”). The criteria for “possible AMI” are based on the recommendations of the WHO MONICA protocol algorithms regarding clinical presentation (“definite clinic”), cardiac enzyme levels changes (“possible elevation of cardiac enzymes”), and ischemic ECG changes (“possible ischemic ECG changes”) in non-fatal events and clinical investigation data (“possible ischemic ECG changes”), and autopsy data (“coronary stenosis > 50%, scar after having had AMI”) and outpatient documentation (“IHD in anamnesis”) in fatal events [28].

### 2.2. Ascertainment of Outcome Events

Information on the causes of death of 25- to 64-year-old inhabitants of Kaunas who died during 2000–2023 was obtained from the Kaunas Civil Registry Office and the Lithuanian Cause of Death and its Consequences Register [30]. All medical death certificates were reviewed to verify the diagnosis. During 2000–2023, causes of death were coded using the ICD-10 [31]. The death certificates with the diagnosis of IHD were selected for verification (codes I20–I25). Following the WHO recommendations, to ascertain all cases of death, medical death certificates that stated the following causes of death were selected: diabetes mellitus (codes E10–E14), obesity (codes E65–E68), dyslipidaemia (codes E78), arterial hypertension (codes I10–I15), other heart diseases (codes I30–I52), cerebral vascular damages (codes I60–I69), diseases of arteries, arterioles, and capillaries (codes I70–I77), and unclear causes of death (codes R95–R99).

### 2.3. Statistical Analysis

The morbidity and mortality rates were calculated per 100,000 population using the world standard [32]. The changes in morbidity and mortality rates were calculated and assessed using the linear regression analysis method using the Joinpoint regression analysis tool [33], and changes are presented as an annual percentage change. Joinpoint regression analyses for 25–64 years and a subgroup (25–54 and 55–64) were performed to assess changes in the annual sex-specific morbidity and mortality rates over the study period. Joinpoint analysis is a data-driven statistical technique that identifies inflection points in the data and fits various linear regression lines based on a pre-selected number of junction points [34]. Since there were 24 points in total, a maximum of 4 junction points were selected for analysis. Linear fits were fitted to the data based on the largest number of junction points. Initially, a zero junction point model was chosen. The actual number of junction points (between 0 and 4) was determined by performing a permutation test. A two-sided *p*-value < 0.05 was considered statistically significant.

### 2.4. Ethical Approval

The study was approved by the Lithuanian Bioethics Committee (ref. No. 14-27/03 December 2001) and the Kaunas Regional Biomedical Research Ethics Committee (ref. No. BE-2-39/19 April 2021). All patient records/information were anonymized and de-identified before the analysis.

## 3. Results

The age-standardized AMI morbidity trends are presented in Figure 1. It was found that the age-standardized morbidity of MI rates in males was, on average, three–four times higher than in females. During 2000–2023, it was observed that in the 25–64-year-old male and female population age-standardized AMI morbidity significantly decreased by 1.3%/yr. (*p* = 0.006) and 2.3%/yr. (*p* < 0.001), respectively.

The trends in the morbidity of AMI by age group are presented in Table 1. It was found that during 2000–2023 the average morbidity of AMI rate in the 25–54-year-old male group was 187.2/100,000 population, while the average morbidity of AMI rate in the 55–64-year-old male group was more than seven times higher and reached 1351.5/100,000 population. The morbidity rate of AMI in females in the youngest (25–54-year-old) age group was almost 6 times lower compared to the corresponding age group of males and amounted to 32.0/100,000 population, and the average morbidity of AMI in females aged 55–64 was 337.8/100,000 population which was more than 10 times higher than in younger females but 4 times lower than in males of the corresponding age. When assessing the trends in the morbidity of AMI in males aged 25–54, a significant decrease was found by an average of 2.2% per year (*p* < 0.001), whilst among older males (55–64 years) we observed only a decreasing trend by an average of 0.7% per year (*p* = 0.1). When assessing the trends in the morbidity of AMI in females over the past 24 years, we found a significant decrease in the morbidity of AMI in both the younger and older groups of females by 2.1% (*p* = 0.002) and 2.4% (*p* < 0.001) per year, respectively.

The trends in the morbidity of AMI by sex and age groups and by Joinpoint regression analysis are presented in Table 2. When assessing the changes in the morbidity of AMI in males aged 25–64 over a one-point period, it was found that only in the 2017–2023 period did the overall incidence of AMI decrease significantly by an average of 7.1% per year. In females aged 25–64 years, the morbidity of AMI significantly decreased on average by 3.8% per year during the 2006–2023 period. Among males aged 25–54, the morbidity of AMI significantly decreased between 2018 and 2023, and among females it decreased during 2009–2023 by 11.7% (95% CI −21.1; −1.0) and by 3.9% (95% CI −6.6; −1.0), respectively. When assessing older males’ and females’ data, similar significant decreasing trends were identified but in a slightly lower percentage for males and very similar to that for females, as in younger age groups. The decreases were on average 5.6% per year and 3.9% per year, respectively.

The age-standardized mortality from IHD is presented in Figure 2. It was found that the mortality from IHD rates in males was, on average, three–four times higher than in females. During 2000–2023, it was observed that in the 25–64-year-old male population age-standardized mortality from IHD significantly decreased (−2.0%/yr., *p* < 0.001), whereas in the female population it was without significant changes.

The trends in the mortality from IHD by age group are presented in Table 3. It was found that in the past 24 years, the average mortality from IHD rate in the 25–54-year-old male group was 44.5/100,000 population, while the average mortality from IHD rate in the 55–64-year-old male group was more than 10 times higher and was 458.8/100,000 population. The mortality from IHD in females in the youngest (25–54-year-old) age group was almost 8 times lower compared to the corresponding age group of males and was 5.3/100,000 population, and the average mortality from IHD rate in females aged 55–64 was 74.3/100,000 population and was more than 14 times higher than in younger females but 6 times lower than in males of the corresponding age. When assessing the trends in the mortality from IHD in males aged 25–54, a significant declining trend was found by an average of 3.3% per year (*p* = 0.002), while among older males (55–64 years) we observed a smaller but significantly declining trend (−1.2%/yr., *p* = 0.005). We did not find significant trends in both age groups (25–54 and 55–64) when assessing the trends in the mortality from IHD in females during 2000–2023.

The trends in the mortality from IHD by sex and age groups and by Joinpoint regression analysis are presented in Table 4. When assessing the changes in the mortality from IHD in males aged 25–64 over a one-point period, it was found that in the 2000–2006-year period the mortality from IHD significantly increased on average by 5.5% per year, then during 2006–2023 period the mortality from IHD decreased significantly by an average of 3.6% per year. In females aged 25–64 years, the mortality from IHD over a one-point period (2000–2004 and 2004–2023) did not change significantly. Among males aged 25–54, the mortality from IHD significantly decreased between 2006 and 2023 by on average 5.8% per year, but among females mortality from IHD did not change significantly among the assessed data when using one-point periods. When assessing older males and females data, similar significant decreasing trends were identified only among males. From 2000 to 2006, the mortality from IHD significantly increased on average by 4.7% per year and it significantly decreased during 2006–2023 by 2.5% per year on average. Mortality from IHD among older females (aged 55–64 years) did not change significantly over a similar period.

## 4. Discussion

The study indicates that during the past two decades (2000–2023) the morbidity of AMI in middle-aged (25–64 years) males and females decreased significantly. When evaluating the changes in the morbidity of AMI in males by age group, a significant decreasing trend was found only in the younger (25–54) age group, while among males in the older (55–64) age group, we did not find a significant change. During the corresponding period, the morbidity of AMI in females in both younger and older age groups also decreased significantly.

The mortality rate from IHD decreased significantly in males, but in females this did not change significantly. When assessing changes in male mortality from IHD across age groups, a significant decrease was found in the two male age groups of 25–54 and 55–64 years; meanwhile, in the mortality from IHD among females across age groups, no significant changes were found over the past twenty years.

The morbidity of AMI is decreasing more rapidly in the female population than in the male population, which may be related to better disease prevention, early diagnosis, or other factors. Mortality from IHD is decreasing more among the male population, which may indicate better treatment or more effective interventions for males compared with females. Statistically significant decreases in middle-aged male mortality from IHD and statistically insignificant female mortality from IHD changes indicate that there are no clear patterns in these groups and that the decrease is not strong enough to be considered significant.

Over the past two decades, the risk profile of the Lithuanian population for CVD has undergone significant changes. Since 2006, Lithuania has implemented a National Screening Program for high-risk individuals (40–54-year-old men and 50–64-year-old women) that has led to a decrease in the prevalence of risk factors for CVD. During this period, the prevalence of arterial hypertension (AH) and dyslipidaemia in the Lithuanian population tended to decrease, especially in females [35,36,37], and a significant decrease in the prevalence of metabolic syndrome has been observed among females [38]. Smoking prevalence has also decreased, especially among males, but has remained stable or even increased among Lithuanian females [39]. According to our study data, over a similar period (since the mid-2000s) the morbidity of AMI in Lithuania has been significantly decreasing in both younger males (25–54 years old) and older females (55–64 years old). The measures to control harmful lifestyle factors (alcohol) adopted in Lithuania may also have led to certain trends in decreasing morbidity of AMI and mortality from IHD, especially in the male population since 2017 [40]. Previous studies have confirmed that CVD prevention programs can reduce the morbidity of AMI and mortality from IHD rates in the long term [41,42,43].

In addition, over the past two decades, significant efforts have been made to improve the diagnosis, management, and treatment of AMI, including outpatient and inpatient care [24]. Nowadays, cardiac reperfusion rates, used as a measure of the effectiveness of AMI treatment in Lithuanian hospitals, are similar to those in Western and Northern European countries [44]. All of the above measures may have contributed to the downward trend in IHD mortality. Unfortunately, no significant trends in mortality from IHD were identified in the Lithuanian middle-aged female population during either the entire period under study or in individual periods. This may be related to certain accessibility and management deficiencies, unclear clinical course of the disease, or other serious concomitant metabolic, respiratory, or renal diseases among females. The stability of female mortality from IHD could be explained by a different clinical picture due to the presence of comorbidities, somewhat delayed trends in access to healthcare facilities, and, as a result, different aspects of treatment that end in a fatal outcome.

In assessing the morbidity of AMI, we observed a continuous downward trend throughout the study period, consistent with other recent studies from some EU countries [3,14,16,18]. In addition, the morbidity rates of AMI presented here clearly confirm that AMI is an important cause of death in the working-age population of both Lithuania and Europe, not only in terms of the numbers but also in terms of proportional mortality, especially in males. It should be noted that despite the decreasing morbidity of AMI and mortality from IHD trends in men and women, the aforementioned indicators in the Kaunas (Lithuania) middle-aged urban population remain quite high compared to other Western and Northern European countries and even some Central and Eastern European countries [26]. Some Eastern European countries, such as Poland, Romania, and Czech Republic, have made progress in reducing AMI morbidity and IHD mortality among females, but significant disparities persist. Lithuania continues to have the highest mortality rates (from two to four times), despite improvements. Poland has achieved the most substantial reductions, likely due to comprehensive prevention and treatment strategies. These improvements are attributed to enhanced primary prevention measures, increased use of statins, and better access to acute coronary care. Romania and the Czech Republic have also made notable progress, but still face challenges in lowering mortality from IHD rates to levels seen in Western Europe. Addressing these disparities requires continued investment in public health initiatives, improved access to quality healthcare, and targeted interventions to reduce cardiovascular risk factors among females in Lithuania.

There are several possible explanations for the observed rates and trends. First, the incidence of AMI has decreased in various populations since the middle of the last century [18,45] due to a decrease in the prevalence of atherosclerotic CVD risk factors and the implementation of modern primary and secondary CVD prevention strategies [46,47]. In recent years, the prevalence of smoking has decreased significantly and the control and management of various CVD risk factors, such as AH, diabetes mellitus, and dyslipidaemia, has intensified [25,48,49]. Modern treatment of IHD has helped to reduce the incidence of AMI complications, which has had direct consequences for the positive trends in IHD mortality [18]. However, arterial blood pressure control remains unsatisfactory, especially in individuals at very high risk of CVD. At the same time, most EU member states at high and very high risk of CVD have not achieved the recommended cholesterol targets in either primary or secondary prevention [23,24,25,50].

In addition, the widespread use of sensitive cardiac troponin assays in AMI diagnostics in recent years has significantly improved the diagnosis of AMI, allowing the detection of both minor coronary artery disease and potential AMI [51,52]. One promising approach is the integration of novel biomarkers such as high-sensitivity troponins, NT-proBNP, inflammatory markers such as hs-CRP or IL-6, and levels of vitamin D-binding protein to detect subclinical myocardial damage and tailor treatment earlier [53].

The lower mortality from the IHD rate may also be partly explained by the increasing incidence of AMI and mortality from other CVDs in patients with chronic CVD pathology, especially in the elderly and other comorbidities [54]. In addition, the prevalence of non-CVDs, such as chronic diseases of the renal or respiratory systems, is increasing [55,56]. These are independent predictors of poor outcomes in AMI and could worsen the survival prognosis of individuals with AMI by worsening kidney and lung function and leading to a higher risk of mortality in both Lithuanian and some European populations. At the same time, strengthening post-AMI care modalities, including telemonitoring, cardiac rehabilitation, and follow-up, could improve long-term outcomes. These approaches should be incorporated into a multidisciplinary, data-driven model of care that accounts for and anticipates sex- and age-specific nuances. Ultimately, translating epidemiological insights into targeted clinical protocols and health policy reforms will be critical for improving cardiovascular resilience in Lithuania and similar healthcare systems.

Without a doubt, with the decreasing mortality rate from IHD the use of the latest and most advanced coronary artery bypass grafting techniques, which can also be used to treat lower-risk cases, especially in patients with multiple comorbidities, could impact the declining trend [54].

It is noteworthy that our observations revealed a greater reduction in morbidity of AMI and mortality from IHD in the younger age group (25–54 years) compared to the older (55–64 years) age groups. This allows us to conclude that there is no progress in the field of CVD prevention in people 55 years and older, which indicates a high prevalence of CVD risk factors in this group. It is well known that the accumulation of harmful lifestyle and clinical factors in CVD begins at an early age [57]. Therefore, the lower reduction in IHD mortality in people over 55 years of age may be associated with the increasing prevalence of CVD risk factors, such as AH, obesity, or diabetes [58]. If appropriate health policies and preventive actions are not implemented, CVD risk factors will soon contribute to premature IHD mortality. Our study also confirmed the stable trends in the mortality from IHD in females over the past years, as in other previously described studies [59]. The main reasons are that females tend to have a decade-later onset of the disease than males and that there are different trends in access and treatment for females compared to males. The under-representation of females in clinical trials has resulted in a lack of published studies disaggregated by sex, highlighting the need for further research in this area [60,61]. Gender-based medicine recognizes that biological sex and gender identity influence health outcomes, disease presentation, diagnosis accuracy, and treatment efficacy. In CVD, females often present with atypical symptoms of AMI (e.g., fatigue, nausea, shortness of breath rather than chest pain), leading to delayed diagnoses. Hormonal, anatomical, and metabolic differences affect plaque formation, clotting risk, and responsiveness to medications. Incorporating a gender-based lens into CVD epidemiology can improve diagnostic accuracy, reduce mortality, and guide policy and healthcare training worldwide. Countries with comprehensive prevention programs, strong gender-specific healthcare training, and rapid emergency response systems (like France or Poland) have seen much greater declines in female AMI morbidity and mortality from IHD trends [62].

Our study results may also have been partly influenced by the COVID-19 pandemic that began in 2020. The COVID-19 pandemic has had a significant impact on AMI morbidity and mortality from IHD worldwide, and Lithuania is no exception. In the early stages of the pandemic (especially in 2020), Lithuania experienced a significant decrease in AMI hospitalizations, consistent with trends across Europe. This decrease is likely due to fear of infection (patients delayed or avoided seeking care due to concerns about the impact of COVID-19 in healthcare settings) and limited access to healthcare (lockdowns, overcrowded hospitals, and reduced availability of elective and non-urgent services reduced the number of presentations). In Lithuania, as in other countries, delays in AMI diagnosis were more common, which was associated with larger AMI sizes, higher rates of complications such as cardiogenic shock or heart failure, and increased in-hospital mortality without urgent coronary reperfusion, as well as limited emergency care capacity as COVID-19 cases strained hospital resources and staff shortages or the redeployment of cardiology staff to COVID care. The impact of the pandemic on preventive care and risk factor control disrupted the usual CVD risk assessment and management, such as hypertension and diabetes control, smoking cessation programs, and lipid control. These disruptions may have worsened CVD risk profiles and may have had long-term consequences for the morbidity of AMI after the acute phases of the pandemic. According to our study, both the morbidity of AMI and mortality from IHD were reduced by a third during that period. In this regard, the COVID-19 pandemic may have had a different impact on changes in IHD mortality. Firstly, SARS-CoV-2 infection may have been a competing cause of death. In addition, the pandemic may have reduced the access of individuals with AMI to healthcare facilities due to restrictions, and therefore, their deaths may have been undercounted. Finally, the decline in IHD mortality during this period may have been due to the uneven impact of COVID-19 on hospitalizations for AMI. Similar effects of the COVID-19 pandemic on IHD mortality have been observed in other EU countries for similar reasons [63,64].

The “post-COVID-19” era, especially in the context of CVD and AMI, is determined by the legacy of the pandemic and new adaptations of healthcare systems. In Lithuania, as elsewhere, these changes are changing trends in morbidity, treatment, and outcomes. In Lithuania, the following emerging trends can be observed in the post-COVID-19 era: (1) In Lithuania and many other countries, hospitalizations for AMI have started to return to pre-pandemic levels, but not completely. Some studies show a persistent underreporting of AMI cases, suggesting that some events may have gone unnoticed or unrecorded during the pandemic. (2) The COVID-19 pandemic has increased physical inactivity, with concomitant weight gain and mental health problems, all of which increase the risk of cardiovascular disease (CVD). Due to disruptions in routine preventive check-ups, follow-ups, and discontinuation of medication, the condition of many patients with CVD has recently deteriorated. (3) Long-term consequences of COVID-19 may contribute to an increased risk of CVD, including AMI, due to prolonged inflammation, endothelial dysfunction, or thrombogenic effects, and as the Lithuanian population ages, this burden may be felt at a higher level over time. (4) One of the positive developments is the increase in telecardiology consultations, remote monitoring, and e-prescriptions. In addition, Lithuania’s e-health infrastructure has expanded, improving accessibility in rural areas (if gaps in digital literacy and infrastructure are addressed). (5) The pandemic has encouraged investment in public health infrastructure and emergency response planning. Lithuania, as an EU member, benefits from regional resilience funding, which can improve CVD care in areas where funds are insufficient.

Comparing some EU healthcare systems, the Lithuanian healthcare system, which provides universal coverage under a compulsory health insurance model, differs significantly from the resource-intensive systems in many Western and Northern European countries. In the area of IHD care, Lithuania has made significant progress in acute care, including an established percutaneous coronary intervention (PCI) network, but disparities remain. Compared to countries such as Germany, Sweden, or the Netherlands, Lithuania faces challenges in areas such as early diagnosis, availability of outpatient cardiac rehabilitation, and rural healthcare infrastructure. Furthermore, Lithuania’s per capita healthcare expenditure is significantly lower, which may contribute to limitations in preventive programs and follow-up care [65]. These systemic differences are important in explaining trends in AMI morbidity and mortality, particularly during and after the COVID-19 pandemic, as resource availability and continuity of care may have played a different role across Europe.

### Strengths and Limitations of the Study

Our study has several limitations. First, it is a retrospective analysis, which depends on the accuracy of the data provided in medical histories and outpatient cards describing the history of the disease and life and describing clinical data and applied diagnostic procedures, especially in cases that had a rapid course and ended fatally, which limited our results.

All cases of AMI were collected retrospectively by the “cold pursuit method”, i.e., after the person left the hospital alive, whilst if the case ended fatally then autopsy or clinical course data were used, with the latter used if an autopsy was not performed. Cases of AMI that ended fatally after leaving the hospital were registered by reviewing the AMI register database in the National Cause of Death Registry database, including during the COVID-19 pandemic. In this study, we did not compare AMI recording by type (non-ST elevation vs. ST elevation), which limited the evaluation of our results. The MONICA project recommendations do not include trends in detailed treatment aspects of AMI cases, which could explain the decreasing mortality from IHD, causing our results to be limited. In addition, the accuracy of the study results could have been determined by incorrect coding of AMI deaths. Indeed, we cannot exclude the possibility that our results did not show some other increasing trends in this period due to incorrect coding, misclassification, or misdiagnosis, especially in individuals with SARS-CoV-2 infection. Future studies will compare changes in AMI morbidity and IHD mortality rates in more detail and identify possible causes in the pre-COVID-19 and post-COVID-19 periods. Similarly, the diagnosis of AMI could have been misleading, especially if the death occurred suddenly outside the hospital, since autopsies are not always performed and diagnoses are based on previous health documentation. In general, the percentage of autopsies in Lithuania has decreased by two–three times over the past decade, which has limited the possibility of adjusting the diagnosis. Therefore, we cannot exclude the possibility that the inaccuracy of the death certificate could have undermined the study’s conclusions. Indeed, death certificate errors contribute to unreliable results and include ICD-10 coding, thereby affecting mortality statistics. However, previous studies have reported high sensitivity values for AMI coding, ranging from 86.0% to 94.0% [66]. Finally, given that previous studies have reported that death certificates may underestimate CVD mortality in younger individuals [67], our results may underestimate the true mortality in younger age groups.

## 5. Conclusions

During the study period (2000–2023) morbidity of AMI among Kaunas middle-aged males and females significantly decreased. The morbidity of AMI in males significantly decreased only in the younger (25–54) age group, while among the older (55–64) age group only a tendency to decrease was observed. During the corresponding period, the morbidity of AMI in females in both younger and older age groups also significantly decreased. The mortality from IHD among Kaunas middle-aged males decreased significantly, but among females there no significant changes. When assessing changes in male mortality from IHD across age groups, a significant decrease was found in both male age groups, 25–54 and 55–64 years, but in females no significant changes were found over the past 25 years.

## Figures and Tables

**Figure 1 medicina-61-00910-f001:**
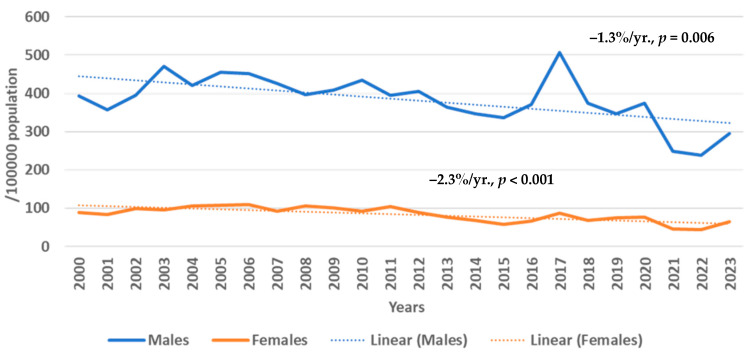
Trends in age-standardized morbidity of acute myocardial infarction in Kaunas (Lithuania) males and females aged 25–64 years from 2000 to 2023 by Joinpoint regression analysis (0 Joinpoints).

**Figure 2 medicina-61-00910-f002:**
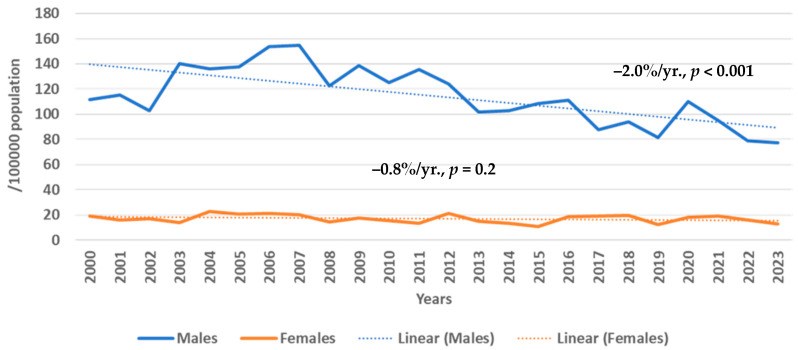
Trends in age-standardized mortality from ischemic heart disease in Kaunas (Lithuania) males and females aged 25–64 years from 2000 to 2023 by Joinpoint regression analysis (0 Joinpoints).

**Table 1 medicina-61-00910-t001:** The rates (per 100,000 population) and trends in morbidity of acute myocardial infarction in Kaunas (Lithuania) males and females by age groups from 2000 to 2023 by Joinpoint regression analysis (0 Joinpoints).

Years	Males		Females	
	25–54	55–64	25–54	55–64
2000	223.39	1229.00	36.93	346.93
2001	172.46	1261.27	35.48	320.03
2002	216.48	1272.19	39.42	393.93
2003	260.85	1506.13	37.47	379.78
2004	208.27	1466.77	33.70	462.18
2005	226.66	1579.75	35.60	460.55
2006	215.10	1610.81	45.15	428.38
2007	233.96	1365.40	28.48	405.45
2008	198.12	1376.67	42.82	416.34
2009	182.97	1517.68	45.15	379.01
2010	215.69	1512.98	26.97	409.07
2011	199.34	1353.45	44.64	396.48
2012	181.55	1501.62	29.16	383.85
2013	189.57	1222.27	32.82	289.67
2014	166.35	1230.81	25.06	284.39
2015	169.19	1171.35	21.77	235.32
2016	178.20	1321.03	24.20	277.99
2017	232.60	1859.71	34.93	346.85
2018	184.31	1311.74	25.68	279.91
2019	152.79	1302.63	26.30	318.90
2020	177.35	1343.02	32.77	293.98
2021	99.46	983.15	20.10	167.92
2022	92.34	954.97	16.87	178.37
2023	115.58	1181.57	27.40	251.36
Average	187.19	1351.50	32.04	337.78
AAPC ^1^ and *p*-value	−2.2%/yr., *p* < 0.001	−0.7%/yr., *p* = 0.1	−2.1%/yr., *p* = 0.002	−2.4%/yr., *p* < 0.001

^1^ AAPC—Average annual percent change.

**Table 2 medicina-61-00910-t002:** The trends in age-standardized morbidity of acute myocardial infarction in the Kaunas (Lithuania) population by sex and age groups during 2000–2023 by the Joinpoint regression analysis.

Age Groups	Sex	Joinpoints (Years)	Period 1	Annual Percentage Change with 95% CI	Period 2	Annual Percentage Change with 95% CI
25–64	Males	2017	2000–2017	−0.2 (+0.9; −0.4)	2017–2023	−7.1 (−12.2; −1.7) *
	Females	2006	2000–2006	+3.7 (−2.2; 10.1)	2006–2023	−3.8 (−5.2; −2.4) *
25–54	Males	2018	2000–2018	−1.1 (−2.3; +0.2)	2018–2023	−11.7 (−21.1; −1.0) *
	Females	2009	2000–2009	+0.6 (−3.7; +5.1)	2009–2023	−3.9 (−6.6; −1.0) *
55–64	Males	2017	2000–2017	+0.3 (−0.8; +1.5)	2017–2023	−5.6 (−10.8; −0.2) *
	Females	2005	2000–2005	+6.9 (−1.5; +16.0)	2005–2023	−3.9 (−5.2; −2.6) *

* *p* < 0.05.

**Table 3 medicina-61-00910-t003:** The rates (per 100,000 population) and trends in mortality from ischemic heart diseases in Kaunas (Lithuania) males and females by age groups from 2000 to 2023 by the Joinpoint regression analysis (0 Joinpoints).

Years	Males		Females	
	25–54	55–64	25–54	55–64
2000	52.34	402.51	4.86	87.69
2001	38.23	495.08	8.59	53.34
2002	39.61	414.20	6.39	70.49
2003	64.92	511.97	4.22	62.60
2004	63.30	494.96	3.13	118.72
2005	60.23	518.45	3.12	106.61
2006	63.16	598.65	9.59	77.11
2007	85.99	494.80	6.44	86.27
2008	55.45	452.52	7.73	48.21
2009	51.79	565.92	7.48	66.88
2010	47.49	508.67	2.06	80.04
2011	48.96	561.68	6.94	45.57
2012	39.67	541.13	4.44	103.87
2013	36.00	425.14	3.41	71.30
2014	28.84	466.41	3.54	61.25
2015	39.50	448.06	2.45	52.29
2016	48.68	417.17	3.77	91.22
2017	29.41	374.44	7.64	73.70
2018	43.21	344.87	12.84	51.68
2019	19.28	388.94	0.01	73.26
2020	44.61	433.43	5.78	77.82
2021	36.85	381.11	5.46	86.01
2022	17.29	383.10	4.16	74.67
2023	13.86	388.23	2.76	62.84
Average	44.53	458.81	5.28	74.31
AAPC ^1^ and *p*-value	−3.3%/yr., *p* = 0.002	−1.2%/yr., *p* = 0.005	−0.4%/yr., *p* = 0.8	−0.7%/yr., *p* = 0.4

^1^ AAPC—Average annual percent change.

**Table 4 medicina-61-00910-t004:** The trends in age-standardized mortality from ischemic heart diseases in the Kaunas (Lithuania) population by sex and age groups during 2000–2023 by the Joinpoint regression analysis.

Age Groups	Sex	Joinpoints (Years)	Period 1	Annual Percentage Change with 95% CI	Period 2	Annual Percentage Change with 95% CI
25–64	Males	2006	2000–2006	+5.5 (+1.0; +10.3) *	2006–2023	−3.6 (−4.6; −2.6) *
	Females	2004	2000–2004	+2.9 (−12.4; 20.8)	2004–2023	−1.1 (−2.6; +0.4)
25–54	Males	2006	2000–2006	+7.0 (−3.8; +19.0)	2006–2023	−5.8 (−8.4; −3.1) *
	Females	2020	2000–2020	+0.7 (−3.1; +4.7)	2020–2023	−23.5 (−68.8; +87.4)
55–64	Males	2006	2000–2006	+4.7 (0.0; +9.6) *	2006–2023	−2.5 (−3.4; −1.5) *
	Females	2004	2000–2004	+5.1 (−14.6; +29.5)	2004–2023	−1.2 (−3.1; +0.7)

* *p* < 0.05.

## Data Availability

The original contributions presented in this study are included in the article; further inquiries can be directed to the corresponding authors.

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
