# Peer review of "Trends in Myocardial Infarction Morbidity and Mortality from Ischemic Heart Disease in Middle-Aged Lithuanian Population from 2000 to 2023: Data from Population-Based Kaunas Ischemic Heart Disease Register"

_medicina, 2025, doi:10.3390/medicina61050910_

Round 1
Reviewer 1 Report
Comments and Suggestions for Authors
In this article by Ricardas Radisauskas et al., the authors studied Trends in Myocardial Infarction Morbidity and Mortality from 2 Ischemic Heart Disease in Lithuanian Middle-Aged Population 3 from 2000 to 2023. They found a reduction in the morbidity and mortality among males and a reduction in the morbidity among females. The study showed improved cardiovascular outcomes from 2000 to 2023 due to improved diagnosis, management, and treatment. This study highlights the critical role of diagnosis and disease management in reducing the morbidity and mortality of CVDs and is a source of knowledge for the readers.
Author Response
In this article by Ricardas Radisauskas et al., the authors studied Trends in Myocardial Infarction Morbidity and Mortality from 2 Ischemic Heart Disease in Lithuanian Middle-Aged Population 3 from 2000 to 2023. They found a reduction in the morbidity and mortality among males and a reduction in the morbidity among females. The study showed improved cardiovascular outcomes from 2000 to 2023 due to improved diagnosis, management, and treatment. This study highlights the critical role of diagnosis and disease management in reducing the morbidity and mortality of CVDs and is a source of knowledge for the readers.
Response: Thank you for the response.
Reviewer 2 Report
Comments and Suggestions for Authors
In this observational retrospective study, Dr. Ricardas Radisauskas and colleagues determine and evaluate changes in acute myocardial infarction (AMI) morbidity and mortality from coronary artery disease (CAD) among middle-aged adults in Lithuanian population in the period 2000-2023.
For this purpose, the authors conducted an observational study including data from the Kaunas Ischemic Heart Disease Registry in the previously reported period.
They showed that the age-standardized AMI morbidity significantly decreased both in male
(-1.3%/year) and in female (-2.3%/year). After an age-based sub-analysis, the significant reduction was observed only in the group of males aged 25-54 without differences in the group of males aged 55-64, whereas in females a significant reduction was observed in both age groups.
Furthermore, they found a significant decrease of mortality from CAD in males 25-64, with a consistent reduction in the two age groups of 25-54 and 55-64 years. On contrary, in females no significant changes were found in mortality, nor in one of the two age groups.
The study research topic warrants careful consideration, due to strong interest in changes in epidemiologic data about CAD in a long-term period and in less evaluated populations such as Baltic people, which are considered at very high risk for cardiovascular mortality from WHO.
Overall, this is a quite nicely written article with good statistic methodology.
However, there are some limitations that should be addressed by the authors:
- Major revisions:
- The major issue of the study is related to the retrospective design. Dependence on secondary data introduces a risk of misclassification or incomplete documentation, especially for deaths occurring outside hospitals or during the COVID-19 pandemic. This aspect should be highlighted in the limitations.
- The introduction section seems too short to discuss about CAD morbidity and mortality trends.
- In methods section, to report coronary events or cardiac death the authors should follow the most accepted definitions derived from ESC fourth definition of myocardial infarction (doi:10.1016/j.jacc.2018.08.1038). Please, verify that all the reported events followed these definitions.
- About the evaluation of morbidity and mortality due to AMI, a sub-analysis to evaluate the different outcomes between non-ST-elevation myocardial infarction (NSTEMI) and ST-elevation myocardial infarction (STEMI) should be useful to better understand the advances in the diagnosis and management of these two different types of AMI.
- The authors revealed a trend toward lower mortality in males of all age groups. It could be helpful to report the trends of some progresses in the treatment of AMI such as more intensified antithrombotic therapy, the adoption of intracoronary imaging during primary percutaneous coronary intervention and the use of mechanical circulatory support device to treat cardiogenic shock.
- Along the previous point the authors did not find any change in mortality in females. How the authors explain this aspect? This disparity could be related to different clinical presentation and different management strategies for females. Please, expand this topic.
- In the manuscript it’s not clear the clinical impact of these results. Could the author expand this section, maybe providing a suggestion of what type of intervention is currently required to further reduce the adverse prognosis of the most impactful cardiovascular disease worldwide such as AMI? Perhaps investigating additional biomarkers? Front Biosci (Elite Ed). 2010 Jun 1;2(3):796-804. doi: 10.2741/e140
- Minor revisions:
- In all the text we suggest you modify the term “ischemic heart disease (IHD)” with “coronary artery disease (CAD)” as more accepted in literature globally. Please, reconcile.
- Page 3, line 101: please remove one space before “The study used the…”.
- Page 6, line 196: please remove one space before “When assessing older males…”
- In table 2, please apply a correct formatting for “Male” and “2018” in the same line with other characters.
Author Response
In this observational retrospective study, Dr. Ricardas Radisauskas and colleagues determine and evaluate changes in acute myocardial infarction (AMI) morbidity and mortality from coronary artery disease (CAD) among middle-aged adults in Lithuanian population in the period 2000-2023.
For this purpose, the authors conducted an observational study including data from the Kaunas Ischemic Heart Disease Registry in the previously reported period.
They showed that the age-standardized AMI morbidity significantly decreased both in male
(-1.3%/year) and in female (-2.3%/year). After an age-based sub-analysis, the significant reduction was observed only in the group of males aged 25-54 without differences in the group of males aged 55-64, whereas in females a significant reduction was observed in both age groups.
Furthermore, they found a significant decrease of mortality from CAD in males 25-64, with a consistent reduction in the two age groups of 25-54 and 55-64 years. On contrary, in females no significant changes were found in mortality, nor in one of the two age groups.
The study research topic warrants careful consideration, due to strong interest in changes in epidemiologic data about CAD in a long-term period and in less evaluated populations such as Baltic people, which are considered at very high risk for cardiovascular mortality from WHO.
Overall, this is a quite nicely written article with good statistic methodology.
However, there are some limitations that should be addressed by the authors:
Major revisions:
The major issue of the study is related to the retrospective design. Dependence on secondary data introduces a risk of misclassification or incomplete documentation, especially for deaths occurring outside hospitals or during the COVID-19 pandemic. This aspect should be highlighted in the limitations.
Response: Thank you for the recommendation. We have updated our limitations section with this relevant information. All cases of AMI were collected retrospectively, by "cold pursuit method", e. g., clinical course data were used in cases where the person left the hospital alive, and if the case ended fatally, either autopsy or clinical course data were used, in cases where autopsy was not performed. Cases of AMI that ended fatally after leaving the hospital were registered by reviewing the AMI register database in the National Cause of Death Registry database, including the COVID-19 pandemic period. These potential losses or discrepancies in AMI case recording are indicated in the limitations section.
The introduction section seems too short to discuss about CAD morbidity and mortality trends.
Response: Thank you for the recommendation. We have updated the introduction section with added information about CAD morbidity and mortality trends. During 2012-2020, standardized mortality rates from AMI for males remained unchanged in some Central and Eastern European countries such as Bulgaria, Romania, Slovakia, and Slovenia, while they significantly decreased in Poland, Hungary, the Czech Republic, Latvia, Estonia, and Croatia. During the corresponding period, standardized mortality rates from AMI for females remained unchanged in Bulgaria, Croatia, Slovakia, and Slovenia, while they significantly decreased in Hungary, the Czech Republic, Estonia, Latvia, Poland, and Romania (Zuin M et al., 2023).
In methods section, to report coronary events or cardiac death the authors should follow the most accepted definitions derived from ESC fourth definition of myocardial infarction (doi:10.1016/j.jacc.2018.08.1038). Please, verify that all the reported events followed these definitions.
Response: Thank you for the recommendation. During the study period, all AMI cases were collected and verified according to the recommendations of the WHO MONICA protocol for AMI case collection to allow for a comparison of the morbidity of AMI and mortality from IHD rates.
About the evaluation of morbidity and mortality due to AMI, a sub-analysis to evaluate the different outcomes between non-ST-elevation myocardial infarction (NSTEMI) and ST-elevation myocardial infarction (STEMI) should be useful to better understand the advances in the diagnosis and management of these two different types of AMI.
Response: Thank you for the recommendation. In this study, we did not aim to compare myocardial infarction recording by AMI type (NSTEMI vs STEMI), so we included this information in the study limitations subsection.
The authors revealed a trend toward lower mortality in males of all age groups. It could be helpful to report the trends of some progresses in the treatment of AMI such as more intensified antithrombotic therapy, the adoption of intracoronary imaging during primary percutaneous coronary intervention and the use of mechanical circulatory support device to treat cardiogenic shock.
Response: Thank you for the recommendation. The MONICA project recommendations do not include trends in detailed treatment aspects of AMI cases, but from other clinical trials conducted in Lithuania, modern treatment aspects such as antithrombotic therapy and primary percutaneous coronary intervention have been increasingly performed over the past 10 years, especially among males. This information was also included in the study limitations subsection.
Along the previous point the authors did not find any change in mortality in females. How the authors explain this aspect? This disparity could be related to different clinical presentation and different management strategies for females. Please, expand this topic.
Response: Thank you for the recommendation. Yes, no changes in female mortality were found during the study. The stability of female mortality could be explained by a different clinical picture, due to the presence of comorbidities, atypical symptoms leading to somewhat delayed trends in access to health care facilities, and, as a result, different aspects of treatment that end in a fatal outcome. With these statements, we expanded the discussion part.
In the manuscript it’s not clear the clinical impact of these results. Could the author expand this section, maybe providing a suggestion of what type of intervention is currently required to further reduce the adverse prognosis of the most impactful cardiovascular disease worldwide such as AMI? Perhaps investigating additional biomarkers? Front Biosci (Elite Ed). 2010 Jun 1;2(3):796-804. doi: 10.2741/e140
Response: Thank you for the suggestion. We believe that the inclusion of additional biomarkers, such as C-reactive protein and sensitive TnI indicators, in clinical practice is very important, because these indicators, which are very good predictors of the outcome of AMI, could contribute to the prediction of the possible development of AMI, so that AMI cases can be identified and confirmed as early as possible, and these AMI cases can be brought to the intensive care unit and interventional cardiologists as early as possible. The findings of this study highlight the ongoing burden and vulnerability of AMI care in the face of systemic disruptions such as the COVID-19 pandemic. Clinically, these results highlight the need for more proactive, targeted strategies to reduce the adverse prognosis of AMI, the most important CVD worldwide. Interventions should prioritize early detection and risk stratification, especially in high-risk groups such as females and the elderly. One promising approach is the integration of novel biomarkers such as high-sensitivity troponins, NT-proBNP, inflammatory markers such as hs-CRP or IL-6, and levels of vitamin D-binding protein to detect subclinical myocardial damage and tailor treatment earlier. At the same time, strengthening diagnostics and early interventional treatment approach, timely access to AMI centres, post-AMI care modalities, including telemonitoring, cardiac rehabilitation, and follow-up, could improve long-term outcomes. These approaches should be incorporated into a multidisciplinary, data-driven model of care that accounts for and anticipates sex and age-specific nuances. Ultimately, translating epidemiological insights into targeted clinical protocols and health policy reforms will be critical to improving cardiovascular resilience in Lithuania and similar healthcare systems.
Minor revisions:
In all the text we suggest you modify the term “ischemic heart disease (IHD)” with “coronary artery disease (CAD)” as more accepted in literature globally. Please, reconcile.
Response: Thank you for the suggestion. If possible, we will stick to the term “ischemic heart disease (IHD)”.
Page 3, line 101: please remove one space before “The study used the…”.
Response: Thank you for the suggestion. We have removed the space.
Page 6, line 196: please remove one space before “When assessing older males…”
Response: Thank you for the suggestion. We have removed the space.
In table 2, please apply a correct formatting for “Male” and “2018” in the same line with other characters.
Response: Thank you for the recommendation. In Table 2, we have corrected the formatting for “Male” and “2018”.
Reviewer 3 Report
Comments and Suggestions for Authors
The study examined acute myocardial infarction (AMI) and ischemic heart disease (IHD) over a 23-year period from 2000 to 2023 among residents aged 25 to 64 years living in Kaunas, Lithuania.It provides a detailed analysis of the incidence and mortality rates, and provides very valuable epidemiological data. It is a study that clarifies trends by gender and age group with a consistent methodology over a long period of time, and I think that there is some novelty.
On the other hand, there was a sense that there was room for improvement in several areas. Although the impact of the COVID-19 pandemic is specifically mentioned, there has been no statistically adjusted analysis of the impact of the pandemic, and it is not clear how the impact is reflected in the results. In the future, it is hoped that a clearer perspective will be presented on how the findings of this study can be applied to public health policy in the "post-COVID" era.
In addition, the interpretation of the results for women is somewhat ambiguous, and there seems to be a lack of consideration of gender differences in healthcare, including background and social factors. For an international audience, the significance of this study will be further enhanced by adding a supplementary explanation of gender-based medicine and comparisons with data from other countries.
On page 2, line 55, the authors mention that “The Eastern and Central EU countries have been less frequently studied.” It would strengthen the manuscript to briefly discuss why these regions have been understudied and what barriers (e.g., data availability, healthcare system differences) may have contributed to this gap.
Please elaborate on whether any validation of the statistical models used in this study was conducted. If model validation was performed, please describe the methods (e.g., sensitivity analyses, goodness-of-fit tests) used to confirm their appropriateness.
The definition and diagnostic criteria for “possible AMI” should be more clearly stated. This clarification would improve transparency and enhance the reproducibility of the study.
While the manuscript acknowledges the impact of the COVID-19 pandemic, the discussion could be expanded to address the following:
How the pandemic specifically influenced AMI incidence and mortality in the Lithuanian context;
What distinguishes the “after COVID-19” era, and how might these trends evolve moving forward?
To enhance the manuscript’s relevance to an international audience, it would be helpful to include a brief comparison between Lithuania’s healthcare system and those of Western and Northern European countries, especially in the context of ischemic heart disease care (see p.10, line 309).
On page 9, line 266, the authors note that there were “no significant changes” in female mortality. Further discussion on possible explanations would improve this section. For example, could delayed diagnosis in women due to atypical symptoms or comorbidities such as diabetes mellitus play a role? Exploring this would deepen the manuscript’s insight into sex-specific disparities.
The study excludes individuals aged 65 and older. Please discuss the rationale behind this exclusion and its potential impact on the generalizability of the findings, especially given the high prevalence of IHD in older populations.
As mentioned above, the discussion would benefit from clearer suggestions for future research directions and public health implications. In particular, specific recommendations for targeted interventions for women and older adults would enhance the manuscript's societal value.
Author Response
The study examined acute myocardial infarction (AMI) and ischemic heart disease (IHD) over a 23-year period from 2000 to 2023 among residents aged 25 to 64 years living in Kaunas, Lithuania. It provides a detailed analysis of the incidence and mortality rates, and provides very valuable epidemiological data. It is a study that clarifies trends by gender and age group with a consistent methodology over a long period of time, and I think that there is some novelty.
On the other hand, there was a sense that there was room for improvement in several areas. Although the impact of the COVID-19 pandemic is specifically mentioned, there has been no statistically adjusted analysis of the impact of the pandemic, and it is not clear how the impact is reflected in the results. In the future, it is hoped that a clearer perspective will be presented on how the findings of this study can be applied to public health policy in the "post-COVID" era.
Response: Thank you for the recommendation. The possible impact of the COVID-19 pandemic on changes in AMI morbidity and mortality from IHD is mentioned in our discussion and limitations subsection. Future studies will compare changes in AMI morbidity and IHD mortality in more detail and identify possible causes in the pre-COVID-19 and post-COVID-19 periods.
In addition, the interpretation of the results for women is somewhat ambiguous, and there seems to be a lack of consideration of gender differences in healthcare, including background and social factors. For an international audience, the significance of this study will be further enhanced by adding a supplementary explanation of gender-based medicine and comparisons with data from other countries.
Response: Thank you for the recommendation. We have added a supplementary explanation of gender-based medicine and comparisons with data from other countries. Gender-based medicine recognizes that biological sex and gender identity influence health outcomes, disease presentation, diagnosis accuracy, and treatment efficacy. In cardiovascular disease, females often present with atypical symptoms of myocardial infarction (e.g., fatigue, nausea, shortness of breath rather than chest pain), leading to delayed diagnoses. There is a historical underrepresentation of females in cardiac clinical trials, limiting data for tailored treatment guidelines. Hormonal, anatomical, and metabolic differences affect plaque formation, clotting risk, and responsiveness to medications. Incorporating a gender-based lens into cardiovascular epidemiology can improve diagnostic accuracy, reduce mortality, and guide policy and healthcare training worldwide. Countries with comprehensive prevention programs, strong gender-specific healthcare training, and rapid emergency response systems (like France or Poland) have seen much greater declines in female AMI morbidity and mortality from IHD trends.
On page 2, line 55, the authors mention that “The Eastern and Central EU countries have been less frequently studied.” It would strengthen the manuscript to briefly discuss why these regions have been understudied and what barriers (e.g., data availability, healthcare system differences) may have contributed to this gap.
Response: Thank you for the recommendation. We adding more information and explanation about the differences between the Eastern and Central EU countries. Eastern and Central European countries, including Lithuania, have historically been underrepresented in cardiovascular research, particularly in large-scale multinational studies. Several factors contribute to this gap. One of the main barriers is limited data infrastructure and interoperability, which hinders contributions to or access from large cardiovascular registries. Differences in healthcare system organization, such as differences in rapid emergency response systems, timely advanced interventions (e.g., PCI), and follow-up care models, also complicate cross-country comparisons and discourage standardized data collection. In addition, resource constraints and reduced research funding in some ECE countries have historically limited their participation in international collaborations. As a result, regional patterns of disease presentation, access to care, and outcomes remain poorly characterized. By focusing on Lithuania, this study helps to address this gap and provides a model for future research on context-specific cardiovascular risk and care dynamics in similarly structured healthcare systems.
Please elaborate on whether any validation of the statistical models used in this study was conducted. If model validation was performed, please describe the methods (e.g., sensitivity analyses, goodness-of-fit tests) used to confirm their appropriateness.
Response: Thank you for the recommendation. Model validation by sensitivity analyses, goodness-of-fit tests. The Joinpoint Regression Program, as the main validity criterion, calculates/uses/performs permutation tests and APC/AAPC values.
The definition and diagnostic criteria for “possible AMI” should be more clearly stated. This clarification would improve transparency and enhance the reproducibility of the study.
Response: Thank you for the recommendation. The criteria for “possible AMI” are based on the recommendations of the WHO MONICA protocol algorithms by clinical presentation (“definite clinic”), cardiac enzyme levels changes (“possible elevation of cardiac enzymes”), and ischemic ECG changes (“possible ischemic ECG changes”) in non-fatal events and clinical investigation data (“possible ischemic ECG changes”), autopsy data (“coronary stenosis >50%, scar after having had AMI”) and outpatient documentation (“IHD in anamnesis”) in fatal events) [WHO, MONICA, 1990]. The additional information was included into Materials and Methods section.
While the manuscript acknowledges the impact of the COVID-19 pandemic, the discussion could be expanded to address the following:
How the pandemic specifically influenced AMI incidence and mortality in the Lithuanian context;
Response: Thank you for the question. The COVID-19 pandemic has had a significant impact on AMI morbidity and mortality from IHD worldwide, and Lithuania was no exception. In the early stages of the pandemic (especially in 2020), Lithuania experienced a significant decrease in AMI hospitalizations, consistent with trends across Europe. This decrease is likely due to fear of infection (patients delayed or avoided seeking care due to concerns about the impact of COVID-19 in healthcare settings) and limited access to healthcare (lockdowns, overcrowded hospitals, and reduced availability of elective and non-urgent services reduced the number of presentations). An increase in outpatients and more severe cases due to delayed treatment was reported. In Lithuania, as in other countries, delays in AMI diagnosis were more common, which was associated with larger AMI sizes, higher rates of complications such as cardiogenic shock or heart failure, and increased in-hospital mortality without urgent coronary reperfusion, as well as limited emergency care capacity as COVID-19 cases strained hospital resources and staff shortages or the redeployment of cardiology staff to COVID care. The impact of the pandemic on preventive care and risk factor control disrupted the usual CVD risk assessment and management, such as hypertension and diabetes control, smoking cessation programs, and lipid control. These disruptions may have worsened CVD risk profiles and may have had long-term consequences for the morbidity of AMI after the acute phases of the pandemic. Regional and socioeconomic disparities showed that rural areas and lower-income populations in Lithuania faced greater barriers during the pandemic, further increasing inequalities in AMI outcomes.
What distinguishes the “after COVID-19” era, and how might these trends evolve moving forward?
Response: Thank you for the question. The “post-COVID-19” era, especially in the context of CVD and AMI, is determined by the legacy of the pandemic and new adaptations of healthcare systems. In Lithuania, as elsewhere, these changes are changing trends in morbidity, treatment, and outcomes. In Lithuania, the following emerging trends can be observed in the post-COVID-19 era: 1) In Lithuania and many other countries, hospitalizations for AMI have started to return to pre-pandemic levels, but not completely. Some studies show a persistent underreporting of AMI cases, suggesting that some events may have gone unnoticed or unrecorded during the pandemic; 2) The COVID-19 pandemic has increased physical inactivity, with concomitant weight gain and mental health problems, all of which increase the risk of cardiovascular disease (CVD). Due to disruptions in routine preventive check-ups, follow-up and discontinuation of medication, the condition of many patients with CVD has recently deteriorated; 3) long-term consequences of COVID-19 may contribute to an increased risk of CVD, including AMI, due to prolonged inflammation, endothelial dysfunction or thrombogenic effects, and as the Lithuanian population ages, this burden may be felt at a higher level over time; 4) one of the positive developments: more telecardiology consultations, remote monitoring, e-prescriptions; in addition, Lithuania's e-health infrastructure has expanded, improving accessibility in rural areas (if gaps in digital literacy and infrastructure are addressed); 5) the pandemic has encouraged investment in public health infrastructure and emergency response planning. Lithuania, as an EU member, benefits from regional resilience funding, which can improve CVD care in areas where funds are insufficient.
To enhance the manuscript’s relevance to an international audience, it would be helpful to include a brief comparison between Lithuania’s healthcare system and those of Western and Northern European countries, especially in the context of ischemic heart disease care (see p.10, line 309).
Response: Thank you for the recommendation. Comparing some EU healthcare systems, the Lithuanian healthcare system, which provides universal coverage under a compulsory health insurance model, differs significantly from the resource-intensive systems in many Western and Northern European countries. In the area of IHD care, Lithuania has made significant progress in acute care, including an established percutaneous coronary intervention (PCI) network, but disparities remain. Compared to countries such as Germany, Sweden, or the Netherlands, Lithuania faces challenges in areas such as early diagnosis, availability of outpatient cardiac rehabilitation, and rural healthcare infrastructure, as well as a high prevalence of CVD risk factors. Furthermore, Lithuania’s per capita healthcare expenditure is significantly lower, which may contribute to limitations in preventive programmes and follow-up care. These systemic differences are important in explaining trends in AMI morbidity and mortality, particularly during and after the COVID-19 pandemic, as resource availability and continuity of care may have played a different role across Europe.
On page 9, line 266, the authors note that there were “no significant changes” in female mortality. Further discussion on possible explanations would improve this section. For example, could delayed diagnosis in women due to atypical symptoms or comorbidities such as diabetes mellitus play a role? Exploring this would deepen the manuscript’s insight into sex-specific disparities.
Response: Thank you for the recommendation. The finding that there was “no significant change” in AMI mortality in females during the pandemic warrants further investigation. Females have been found to frequently present with atypical symptoms such as fatigue, dyspnea, or nausea, which may delay diagnosis and treatment. This delay may have been exacerbated during the pandemic, when both patient hesitation and system strain hindered timely care. In addition, women with AMI are more likely to have comorbidities such as diabetes, hypertension, and autoimmune diseases, which complicate clinical assessment and treatment. Despite these challenges, the stability of mortality rates in females may reflect a previous under recognition of AMI in females, suggesting that the pandemic has not significantly altered the trajectory of already suboptimal care. This highlights the need for research into gender-specific public health messages, tailored diagnostic strategies, and gender-specific cardiac care approaches.
The study excludes individuals aged 65 and older. Please discuss the rationale behind this exclusion and its potential impact on the generalizability of the findings, especially given the high prevalence of IHD in older populations.
Response: Thank you for the recommendation. According to the MONICA protocol, data were collected only from 25 to 64 years of age inclusive, and therefore, we did not include persons aged 65 and older in the AMI registry. The reason for their exclusion relates to methodological considerations, such as minimizing confounding by multiple comorbidities, polypharmacy, or frailty, which are more prevalent in older adults and may obscure the effects under study. The exclusion of individuals aged 65 years and older from the study population limits the generalizability of the findings, particularly in the context of IHD, which disproportionately affects older adults. This age group accounts for a significant proportion of AMI cases and is typically characterized by more complex clinical presentations, a higher burden of comorbidities, and an increased risk of mortality. However, this approach also reduces the applicability of the findings to real-world populations, particularly in aging societies such as Lithuania, where the burden of IHD is concentrated in older adults. Future studies will try to include older adults or stratify analyses by age group to more accurately reflect population-level trends and inform age-specific public health strategies.
As mentioned above, the discussion would benefit from clearer suggestions for future research directions and public health implications. In particular, specific recommendations for targeted interventions for women and older adults would enhance the manuscript's societal value.
Response: Thank you for the recommendation. To increase preparedness for future health crises and address persistent disparities in outcomes of IHD, future research should prioritize age- and sex-specific analyses. In particular, studies that focus on the unique symptom profiles and care-seeking behaviours of females are needed to improve early recognition and reduce diagnostic delays. Similarly, given the high prevalence of IHD among older adults, more inclusive study designs that include individuals aged 65 years and older are necessary to fully capture the impact of the pandemic on this vulnerable group. From a public health perspective, targeted interventions should include sex-specific information campaigns to educate females about atypical symptoms of AMI and encourage earlier care-seeking. Improving access to telemonitoring, cardiac rehabilitation, and community-based screening programs for older adults can reduce risk and increase continuity of care. Investing in these targeted strategies could reduce health disparities and strengthen the resilience of cardiovascular care systems in Lithuania.
Reviewer 4 Report
Comments and Suggestions for Authors
The manuscript addresses a significant public health concern by examining long-term trends in acute myocardial infarction (AMI) morbidity and ischemic heart disease (IHD) mortality in a middle-aged Lithuanian population. Drawing on robust registry data, the study offers valuable national-level insights into regional cardiovascular trends. The topic is relevant to a broad readership in cardiovascular epidemiology and public health. Methodologically, the study is well-executed, adhering to established standards such as the WHO MONICA criteria and Joinpoint regression analysis.
However, I have several questions and comments for the authors:
- Although the dataset is solid, the discussion would benefit from a more in-depth contextualization with comparable regional data (e.g., Poland, Romania, Czech Republic). Studies from these populations are cited in the reference list, yet the manuscript does not sufficiently engage with them—particularly when interpreting trends in female mortality.
- I understand that the ages of 25–64 were selected to focus on the economically active population. Monitoring cardiovascular mortality in this demographic is crucial. Nevertheless, it would be equally valuable to assess outcomes in older age groups. Based on the logic of the paper, one might expect similar trends in the 65+ population, driven by the same factors highlighted by the authors. Why was the study not extended to include this older demographic?
- Figure 1 shows a pronounced spike in AMI mortality among men in 2017. What might explain this anomaly? Do the authors have any hypotheses or data that could shed light on this abrupt increase?
Minor Comment:
There are several minor grammatical issues throughout the manuscript that require correction.
Author Response
Reviewer 3
The manuscript addresses a significant public health concern by examining long-term trends in acute myocardial infarction (AMI) morbidity and ischemic heart disease (IHD) mortality in a middle-aged Lithuanian population. Drawing on robust registry data, the study offers valuable national-level insights into regional cardiovascular trends. The topic is relevant to a broad readership in cardiovascular epidemiology and public health. Methodologically, the study is well-executed, adhering to established standards such as the WHO MONICA criteria and Joinpoint regression analysis.
However, I have several questions and comments for the authors:
- Although the dataset is solid, the discussion would benefit from a more in-depth contextualization with comparable regional data (e.g., Poland, Romania, Czech Republic). Studies from these populations are cited in the reference list, yet the manuscript does not sufficiently engage with them—particularly when interpreting trends in female mortality.
Response: Thank you for the recommendation. Some of the Eastern European countries as Poland, Romania, Czech Republic, have made progress in reducing AMI morbidity and IHD mortality among females, but significant disparities persist. Lithuania continues to have the highest mortality rates (from 2 to 4 times), despite improvements. Poland has achieved the most substantial reductions, likely due to comprehensive prevention and treatment strategies. These improvements are attributed to enhanced primary prevention measures, increased use of statins, and better access to acute coronary care. Romania and the Czech Republic have also made notable progress, but still face challenges in lowering mortality from IHD rates to levels seen in Western Europe. Addressing these disparities requires continued investment in public health initiatives, improved access to quality healthcare, and targeted interventions to reduce cardiovascular risk factors among females in Lithuania.
- I understand that the ages of 25–64 were selected to focus on the economically active population. Monitoring cardiovascular mortality in this demographic is crucial. Nevertheless, it would be equally valuable to assess outcomes in older age groups. Based on the logic of the paper, one might expect similar trends in the 65+ population, driven by the same factors highlighted by the authors. Why was the study not extended to include this older demographic?
Response: Thank you for the question. According to the MONICA protocol, data were collected only from 25 to 64 years of age inclusive, and therefore, we did not include persons aged 65 and older in the AMI registry. We have already started collecting data on the incidence of AMI and mortality from IHD in persons aged 65 and over, and in the future, we will be able to compare their trends with those of people aged 25-64.
- Figure 1 shows a pronounced spike in AMI mortality among men in 2017. What might explain this anomaly? Do the authors have any hypotheses or data that could shed light on this abrupt increase?
Response: Thank you for the question. Figure 1 shows a spike in AMI morbidity among Lithuanian men, especially those aged 55–64, in 2017. Several contextual factors suggest plausible causes. In 2017, Lithuania underwent major healthcare and economic reforms that included hospital consolidations, reduced inpatient capacity, and a shift toward outpatient care—changes that may have temporarily disrupted access to timely treatment. Additionally, some healthcare facilities began implementing digital reforms, transitioning to new electronic databases for case registration. This may have artificially increased reported cases due to improved case capture or delayed entries being recorded altogether as a group. Public health policy changes, such as new alcohol control measures, may also have contributed by inducing stress or withdrawal in high-risk individuals. Further compounding the situation, Europe experienced widespread influenza outbreaks in 2017, which are known to trigger cardiovascular events, including AMI, particularly in older adults with pre-existing risk factors. Combined with the already high cardiovascular risk profile of older Lithuanian males, these overlapping factors likely contributed to the observed spike. The additional information was included in the discussion section.
Minor Comment:
There are several minor grammatical issues throughout the manuscript that require correction.
Response: We have corrected previous grammatical errors.
Round 2
Reviewer 2 Report
Comments and Suggestions for Authors
no further comments
Reviewer 3 Report
Comments and Suggestions for Authors
Acceptable